# Reorientation behavior in the helical motility of light-responsive spiral droplets

Federico Lancia [1], Takaki Yamamoto [2], Alexander Ryabchun [1], Tadatsugu Yamaguchi[1], Masaki Sano[3,4]* & Nathalie Katsonis [1]*

The physico-chemical processes supporting life's purposeful movement remain essentially unknown. Self-propelling chiral droplets offer a minimalistic model of swimming cells and, in surfactant-rich water, droplets of chiral nematic liquid crystals follow the threads of a screw. We demonstrate that the geometry of their trajectory is determined by both the number of turns in, and the handedness of, their spiral organization. Using molecular motors as photo-invertible chiral dopants allows converting between right-handed and left-handed trajectories dynamically, and droplets subjected to such an inversion reorient in a direction that is also encoded by the number of spiral turns. This motile behavior stems from dynamic transmission of chirality, from the artificial molecular motors to the liquid crystal in confinement and eventually to the helical trajectory, in analogy with the chirality-operated motion and reorientation of swimming cells and unicellular organisms.

[1] Bio-inspired and Smart Materials, MESA+ Institute for Nanotechnology, University of Twente, PO Box 207, Enschede 7500AE, The Netherlands.
[2] Laboratory for Physical Biology, RIKEN Center for Biosystems Dynamics Research, Kobe 650-0047, Japan. [3] Institute for Natural Sciences, Shanghai Jao Tong University, No. 800 Dongchuan Rd, Minhang District, Shanghai 200240, China. [4] Universal Biology Institute, The University of Tokyo, 7-3-1 Hongo, Bunkyo-ku, Tokyo 113-0033, Japan. *email: sano@phys.s.u-tokyo.ac.jp; n.h.katsonis@utwente.nl

In the search for creating molecular systems with life-like qualities, droplets, proto-cellular constructs, and other small-scale motile entities have been designed and synthesized[1–5] that display chemotaxis[6], phototaxis[7,8], oscillatory motion[9,10], and are able to swim in water with a temperature-responsive velocity[11]. However, these motile systems have yet to demonstrate the sophisticated and purposeful behavior of their living counterparts[12], and, in particular, encoding directionality from the molecular level upwards has remained elusive. The frequently used catalytic decomposition of hydrogen peroxide to achieve propulsion, typically leads to erratic motion, with a notable lack of control over rotational degree of freedom[13]. In contrast, swimming cells and aquatic microorganisms are steered by biomolecular machines and can thus orient their swim. For example, it is the flagellar motor of the *Escherichia coli* bacterium that steers its motion and encodes its tumbling, with chirality a salient feature in this operation[14,15].

Chemists have succeeded in designing and synthesizing a large variety of artificial molecular machines[16–18] and in recent years, the asymmetry and chirality of dynamic and responsive molecules has been harnessed to drive complex macroscopic motion[19], e.g., the twisting of springs[20], the rotation of patterns[21], and the emergence of pulsating vortices[22]. Even so, in all these examples motion sees the center of mass of the object fixed, or in other words the molecular motion fails to convert into motile behavior.

Chirality is key to impart sophisticated motile behavior[23,24]. Movement at small scales requires a shape transformation of the body that breaks a time-reversal invariance[25], and this is arguably why swimming microorganisms must be able to change the shape of their chiral bodies. For example, *E. coli* uses a rotating helical flagellum to swim[14], whereas helix-shaped body of *Helicobacter pylori* and *Campylobacter jejuni* mediates their efficient swimming in challenging media[26]. Besides often relying on chiral body shape transformations, the vast majority of aquatic microorganisms and cells actually swim along helical trajectories, including zooplankton, ciliates, and bacteria[27]. Although the evolutionary advantages of helical motion remain unclear, it appears that the directionality of helical motion is less sensitive to extracellular and intracellular random variations, compared with rectilinear motion[28]. Significantly, swimming in a screw-like fashion offers opportunity for efficient reorientation behaviors set to follow chemical and physical gradients[29,30]. Besides the well-known 'run and tumble' behavior of *E. coli*, these chirality-derived reorientation mechanisms often feature deterministic directional changes, such as in the 'run, reverse, and flick' of *Vibrio alginolyticus*, a marine bacterium that also moves along helical pathways, and for which the flick angle is pre-programmed, i.e., the re-direction does not occur randomly[31,32].

Our general strategy is inspired by the chirality-derived reorientation mechanisms of living systems, where molecular machines drive a change in chiral structures to initiate a reorientation from the molecular level upwards. We have incorporated molecular motors in an (achiral) nematic liquid crystal. The chirality of the motors induces a twist in the liquid crystal and leads to the formation of a light-responsive cholesteric liquid crystal, which is characterized by a helical organization of the molecules (Fig. 1). This liquid crystal was confined into a spherical droplet with perpendicular anchoring—the interface is such that the liquid crystal molecules are oriented perpendicularly to the interface. In these conditions of anchoring, a double spiral disclination line forms at the interface, which reflects the handedness of the cholesteric helix and is associated with a spiral distribution of the director field in the bulk of the droplet[33–36]. Recent work has shown that such chiral droplets follow helical trajectories in surfactant-rich water[37], based on the observation that instabilities can form when micelles solubilize oils from droplets in water[38–40].

Here, we show that photo-inverting the handedness of the molecular motors initiates a cascade of bottom-up events. This molecular-scale event leads to inversion of the liquid crystal helix, further into inversion of the spiral organization of the director field in the droplet, and eventually leading to modification of the flow of fluid matter that steers the helical propulsion. Droplets subjected to such an inversion of handedness reorient their swimming in a direction that is encoded by the number of spiral turns in the droplet.

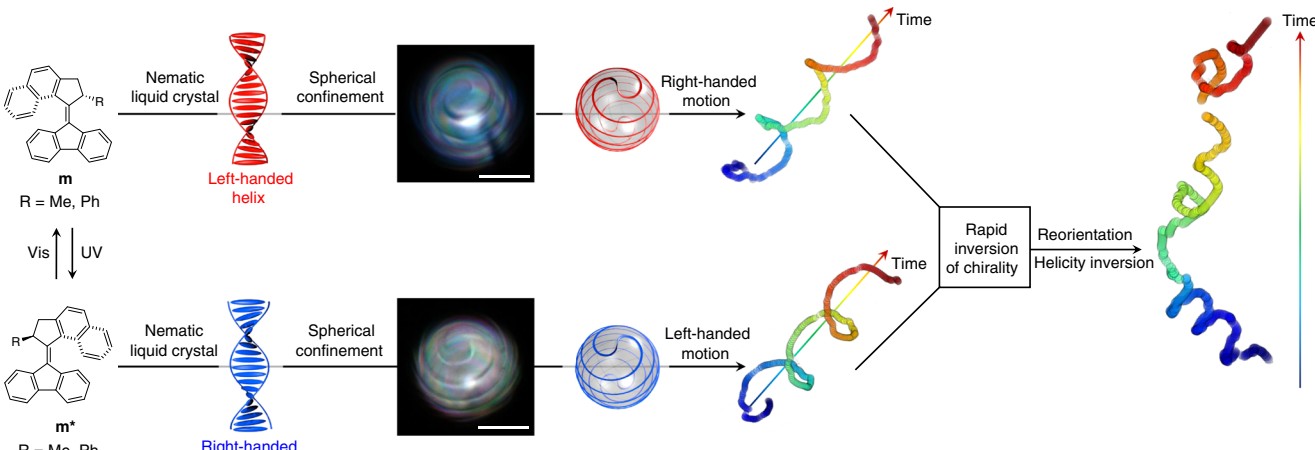

**Fig. 1** Spiral droplets reorient their swim in response to light-induced helix inversion. The molecular chirality of motor **m** and its light-induced isomer **m**\* is transmitted across increasing length scales, from a chiral center in the motor, to the helical shape of the motor and eventually to liquid crystal chirality (left-handed in red, right-handed in blue). In spherical confinement and with the liquid crystal molecules aligning perpendicularly to the interface, a double spiral disclination line forms at the surface of the droplet. This disclination line is visible under polarized optical microscopy (scale bar 50 μm) and is also shown in the model of the droplet (red for left-handed and blue for right-handed). These chiral droplets are propelled along helical trajectories with a handedness that is opposite to that of the droplets. Upon illumination, the molecular motors invert the handedness of their helical shape, in a process that drives their rotation and leads to inversion of liquid crystal handedness. This light-induced inversion is associated with inversion of the helical trajectory and with a deterministic reorientation

## Results

**Helical motion of swimming droplets.** Overcrowded alkenes were used as light-driven molecular rotary motors (Me-**m** and Ph-**m**, Fig. 2). The asymmetric structure of the motors is transmitted to liquid crystals and small concentrations of motor can thus transform an achiral (nematic) liquid crystal into a chiral nematic (cholesteric) liquid crystal that is characterized by a helical organization of the molecules. Molecular motors are unique photo-responsive dopants for liquid crystals because their light-driven operation involves inversion of their helical chirality and thus induces a large-scale, precise, and reversible winding or unwinding of cholesteric helices[41]. These large photo-induced

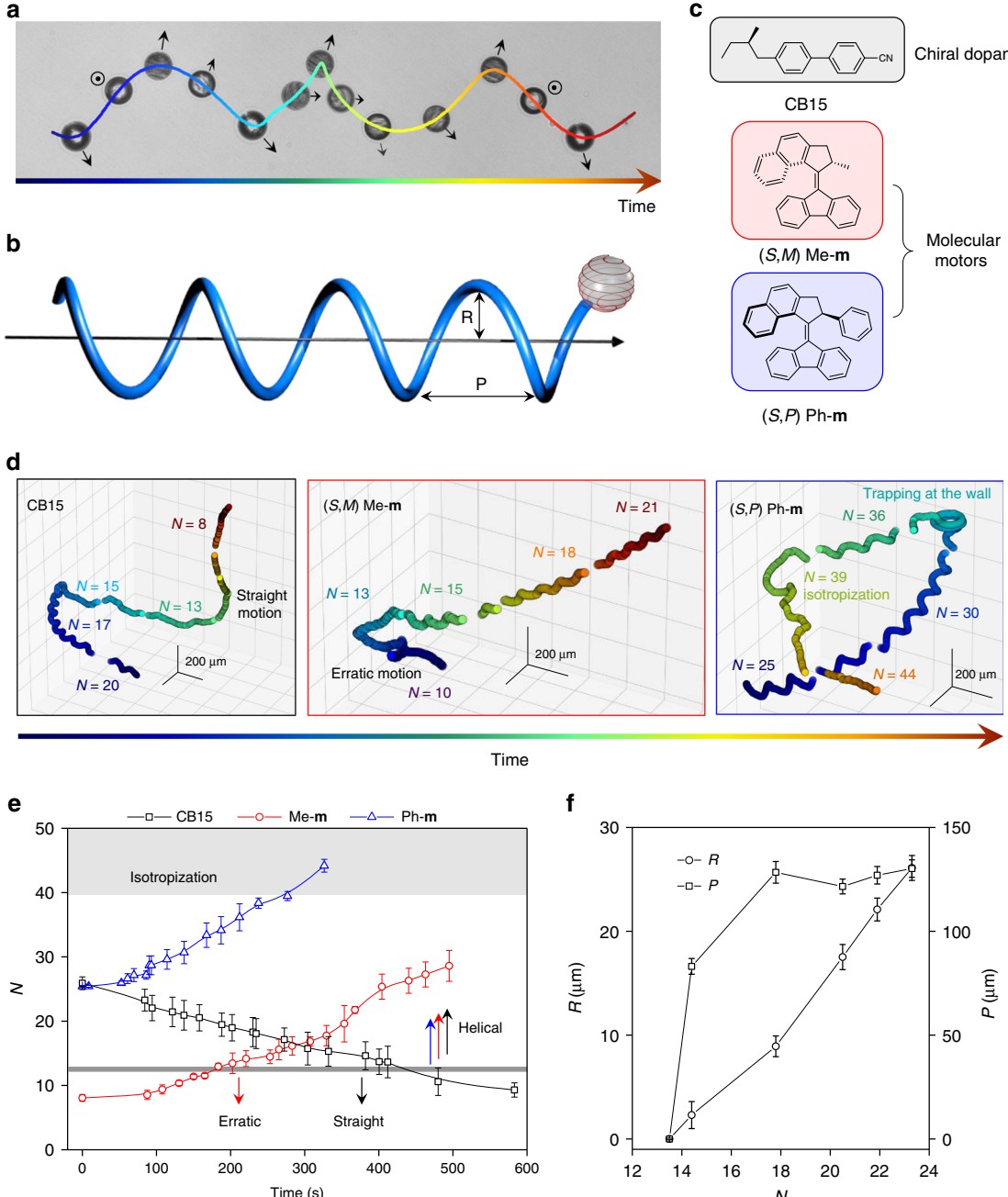

**Fig. 2** Helical motion of spiral droplets. **a** Two-dimensional projection of the trajectory of a chiral droplet. The chirality of the droplet is expressed as a spiral pattern and the axis of the droplet (indicated by a black arrow) undergoes a precession along its helical trajectory. The data acquisition is done by manual z-tracking. **b** Representation of a left-handed droplet moving along a right-handed helical trajectory. The geometry of the motion is characterized by its radius R and pitch P. **c** Chiral molecules used as dopants to induce a cholesteric liquid crystal. CB15 is an enantiomerically pure chiral molecule. Me-**m** and Ph-**m** are molecular motors that modify the pitch of the liquid crystal helix and invert its handedness upon illumination. Ph-**m** induces larger twists than Me-**m**. **d** Complete trajectories for chiral droplets doped with CB15 (left), Me-**m** (center), and Ph-**m** (right). **e** Evolution of the confinement ratio N in time (number of spiral turns on the surface of the droplet), for droplets incorporating chiral dopant CB15 (black square, initial pitch is 1.5 μm), Me-**m** (red circles, initial pitch is 4.3 μm) and Ph-**m** (blue triangle, initial pitch is 1.6 μm). In the case of Me-**m** and Ph-**m**, the concentration of dopant increases over time and eventually the droplet becomes isotropic. The solid lines are guides for the eye. The data are ensemble averaged and error bars correspond to the standard deviation. **f** Dependence of radius R (circles) and pitch P (squares) from the chirality confinement N. The error bars correspond to standard deviation. The data are ensemble averaged for six droplets doped with CB15

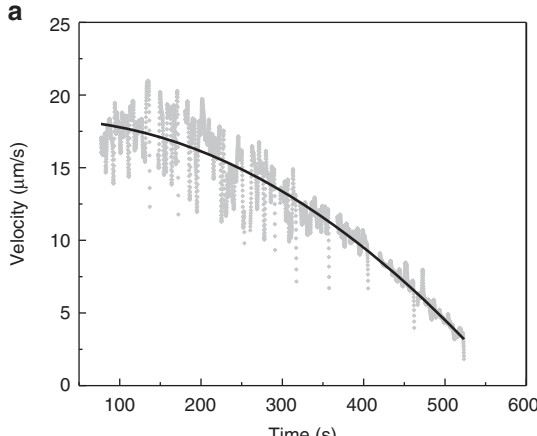
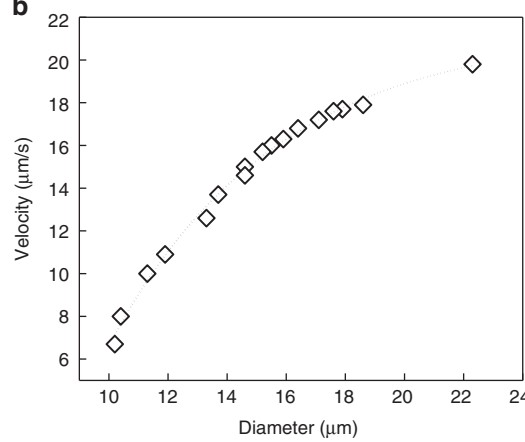

**Fig. 3** Swimming velocity in time and against the size of the droplets. **a** The velocity of a chiral droplet doped with CB15 evolves in time. Here, the initial diameter of the droplet was ca. 19 μm (data shown for one individual droplet). **b** Velocity of motile droplets as a function of their diameter. These data correspond to a collection of drops, and was acquired during their motion

changes in pitch are combined with the possibility to invert helical handedness[42,43].

Droplets with a 20 μm diameter were prepared by injecting the photo-responsive cholesteric liquid crystals in a mixture of water and $D_2O$ enriched in tetradecyl-trimethylammonium bromide surfactant. The size dispersity of the droplets is shown in Supplementary Fig. 1. $D_2O$ was used to match the density of the liquid crystal and thus prevent the droplets from floating or sinking.

Once a droplet is formed, it follows a helical trajectory over several millimeters for up to 8 min. The handedness of the trajectory is constant over time and is always opposite to that of the liquid crystal (Fig. 1). Key to the motion is the presence of a large quantity of surfactants in water, forming micelles that can solubilize small amounts of liquid crystals and thus create inhomogeneities in surface tension that are compensated by a flow that drives the droplets forward, a phenomenon known as the Marangoni effect[38].

**Propulsion mechanism.** The propulsion mechanism can be understood as follows: the solubilization of the liquid crystal by the micelles induces an inhomogeneity of the surfactant at the droplet interface, and consequently an inhomogeneity in the interfacial tension. The resulting gradient in interfacial tension initiates a flow, known as Marangoni flow, which propels the droplet: a larger concentration of surfactant at the front of the droplet combined with a smaller concentration at the rear drag the interfacial fluids from the front to the rear[38,44]. The motion is sustained by a continuous supply of surfactants from the surrounding. Meanwhile, as the surfactant diffuses at the droplet interface, motion ceases when the shrinking droplet is small enough that the diffusion process dominates over the supply of surfactant and the droplet is finally covered by surfactants homogeneously.

Here, the droplets are chiral, and the spiral organization of the molecules in a droplet is specifically associated with the formation of spiral disclination lines (Fig. 1). The molecules are disorganized in these defect lines, and the viscosity is thus lower compared with the interfacial areas where the molecules are oriented homeotropically[45]—in other words the Marangoni flow is faster along the spiral disclination lines. As the entire system of bulk fluid and droplet is torque free, the internal flow consequently rotates, and then the droplet itself spins along a helical trajectory that is characterized by a pitch P and radius R (Fig. 2a, b, and

Supplementary Movie 1). Overall, the Marangoni flow thus couples to the chiral variation of the director via the viscosity[37,46–48]. Symmetry arguments are of general relevance for driving helical motion and also used in theoretical models for the helical propulsion of orthotropic particles[49].

When a droplet reaches the top or bottom boundaries of the chamber, it gets trapped and starts swimming in circles. Although the trapping mechanism is possibly different, a similar behavior is found for microorganisms swimming close to boundaries (for a case of trapping that involves Ph-**m** droplets see Fig. 2d right panel)[50,51]. Hereafter, trapped trajectories and other sections of 2D trajectories, where droplets interact with the boundaries of the chamber, are excluded from the discussion and from any statistical analysis and the data we discuss concerns exclusively free motion and reorientation.

We used light-responsive chiral dopants that behave differently in response to light. The possibility to vary the confinement of the cholesteric helix in the drop allowed investigating droplets that show different swimming behaviors and to unravel the role of chirality in the chirality of their motion. The chirality confinement ratio $N = 2d/p$, where $d$ is the diameter of the liquid crystal droplet and $p$ is the pitch, corresponds to the number of spiral turns in the drop and therefore it can be used to quantify the chiral character of the drop and how it varies during motion.

Liquid crystals doped with $(10.00 \pm 0.01)$ wt% CB15 have an initial pitch of ca. 1.5 μm, with initial values of $N$ ca. 20, and with such a confinement ratio the droplets start swimming helically as soon as they are injected in solution (Fig. 2d). During movement, the number of spiral disclination lines decreases until eventually the droplet becomes a compensated nematic and motion also loses its chiral character (Fig. 2d, left panel). The number of spiral lines decreasing indicates that the pitch of the cholesteric helix increases over time, which we attribute to the empty micelles co-solubilizing CB15 together with the liquid crystal (Supplementary Fig. 2). In contrast, for cholesteric droplets doped with $(0.38 \pm 0.01)$ wt% Me-**m** and having an initial pitch of 4.3 μm $(N = 10)$, Me-**m** concentrates in the droplet during the motion and translates into an increase of spiral turns. This observation is consistent with previous report on chiral dopants structurally more complex than CB15, which have been proven to enrich in shrinking droplets[38].

**Threshold value for the helical swim.** The propulsion of cholesteric droplets doped with either CB15 or Me-**m** offers

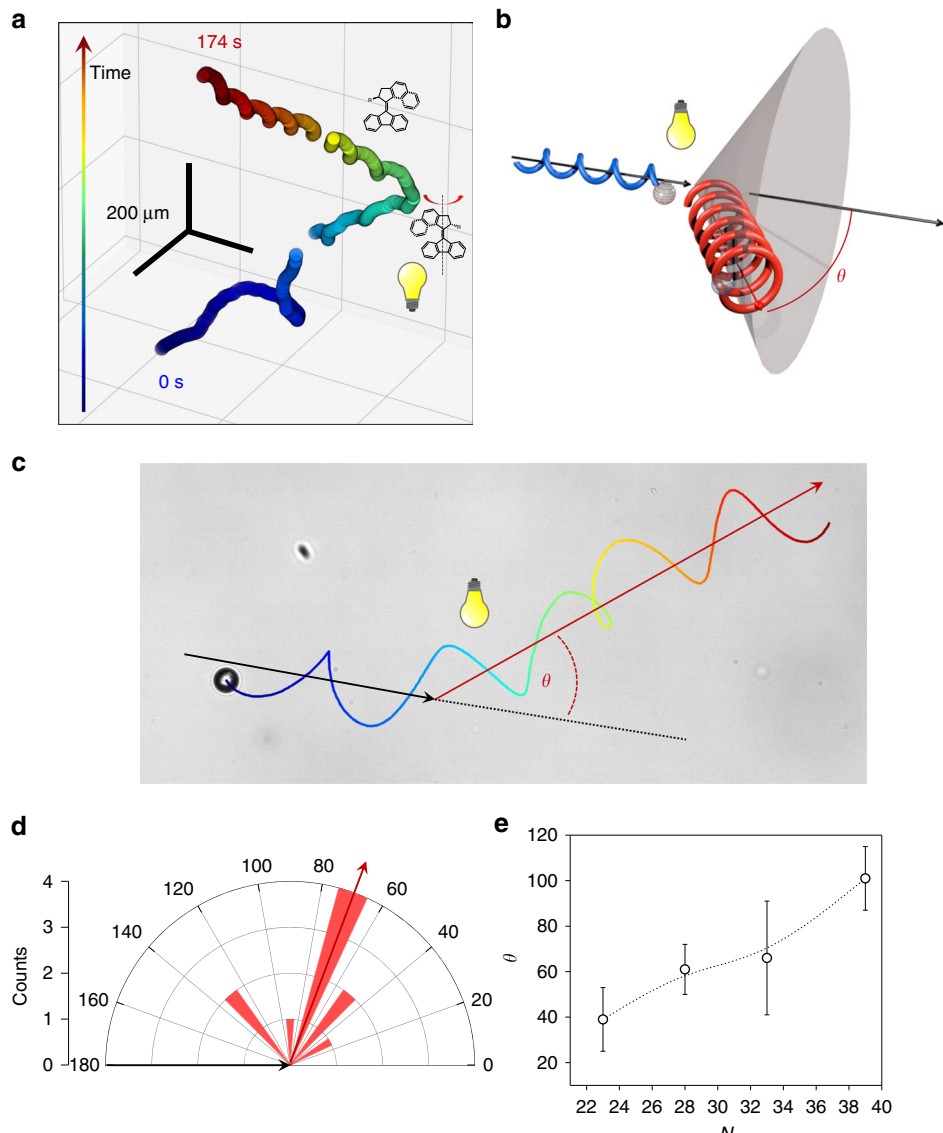

**Fig. 4** Reorientation of spiral droplets in response to light. **a** Three-dimensional motion of a droplet incorporating Me-**m** motor. Upon strong illumination, the helix inversion operates a sharp reorientation. **b** A sharp directional change is operated by the spiral droplets upon helix inversion. A chiral droplet follows a right-handed helical trajectory (in blue) and upon strong illumination helix inversion in the droplet inverts the handedness of the trajectory (in red) and modifies the direction of droplet motion. The direction of reorientation is characterized by the angle $\theta$. **c** Two-dimensional trajectory of a droplet that undergoes directional change upon rapid helix inversion. The angle $\theta$ is defined as the angle between the helical axis of motion before (black arrow) and after illumination (red arrow). **d** Measured reorientation angle $\theta$ for $N = 32$. These angle values are meaningful only between 0 and 180, which means that a cone of deterministic reorientation is associated to each reorientation angle $\theta$. **e**, Dependence of $\theta$ from the chirality confinement ratio $N$. The angle of photo-stimulated re-direction increases with $N$. The error bars indicate standard deviation, the dashed line is a guide for the eye

complementary data that point at a threshold value of $N$ over which helical motion occurs. The helical twisting power of Ph-**m** is larger than that of Me-**m** in the same liquid crystal, and was used as a dopant to form droplets of initial pitch ca. 1.6 μm ($N = 25$, $0.68 \pm 0.01$ wt % of Ph-**m**). These chiral droplets doped by Ph-**m** are characterized by a confinement ratio that is similar to that of the CB15-doped drops, and, similarly, over time, the chiral confinement increases in these droplets, until the high concentration of motor disrupts the liquid crystal order and the droplet turns isotropic (Fig. 2d, right panel). The time evolution of the parameter $N$ for droplets containing each of these three different dopants (Supplementary Fig. 3) points to a threshold value of $N$ above which the chirality of the system is sufficient to impart angular velocity to the droplet and consequently drives the motion along helical paths (Fig. 2e). This threshold value was estimated by comparing propulsion of droplets of CB15 and Me-**m** and was consistently found in the range $12 < N < 14$ (Fig. 2d, e). We note that CB15-doped droplets swim straight below the threshold $N$, whereas Me-**m** doped droplets swim more erratically below that threshold. The difference likely originates in the distortion of the director field caused upon droplet injection in the larger Me-**m** droplets (Supplementary Fig. 3), and the liquid crystalline organization takes longer to emerge in these larger droplets.

Trajectories of CB15-doped droplets were followed in time (Fig. 2f). The correlation of the pitch P and radius R values with the chirality confinement $N$ provides additional evidence that chirality defines the geometry of the helical trajectories. Indeed, the pitch of the helical trajectory increases when the helix gets more confined in the droplet, until it reaches a plateau at $N \approx 18$. The radius of the helical trajectory increases linearly with $N$,

which suggests that increasing the chiral character of the drop increases the angular velocity of the droplet.

Along a helical trajectory, the average velocity of a droplet of 20 μm in diameter is ca. 18 μm/s. In comparison, *E. coli* with a cell body of typically 2–3 μm swims at a velocity of 20–40 μm/s. As mentioned above, however, the droplets are shrinking while they move, and therefore the velocity of a given droplet decreases over time (Fig. 3a). The observation that larger droplets swim faster agrees with trends that were established for isotropic oil drops in water (Fig. 3b)[52]. This relation between velocity and size is related to the asymmetry in the surroundings of a droplet: for the propulsion driven by the Marangoni effect, a front-rear asymmetric distribution of surfactant at the surface of the droplet is required[53]. When the droplet is sufficiently small, the supply of surfactant molecules from the surrounding solution can readily reduce such an inhomogeneous distribution. In contrast, for larger droplets, the supply is less sufficient to reduce the inhomogeneity. Therefore, the speed of the droplet decreases with decreasing droplet size.

**Molecularly encoded reorientation behavior**. Under illumination, the operation of the molecular motors inverts the cholesteric helix in the droplets and induces the same absolute values of pitch. Remarkably, when the handedness of the spiral is forced to invert rapidly, by strong irradiation with light (ca. 1 s for the overall inversion), the droplets reorient significantly in three-dimensional space and the handedness of their trajectories inverts also (Fig. 4a, Supplementary Movie 2, Supplementary Figs. 5–10). We note that helix inversion occurs in one second and moreover, the final pitch is equal to the initial pitch, therefore the confinement ration $N$ does not vary considerably within the timescale of the inversion. Specifically, and considering the time evolution of N (see Fig. 2e), we estimate that $N$ increases of only 0.03 units in one second. This association between photo-induced reorientation and dynamic helix inversion is reminiscent of the 'helical klinotaxis' that is responsible for the complex photo-tactic behavior of *Chlamydomonas reinhardtii*. This unicellular green alga swims along helical trajectories and, in response to light, it inverts the handedness of its helical motion, which makes it swim towards the light. As a protective mechanism, when the light is too intense, *C. reinhardtii* inverts the handedness of its helical trajectory again in order to swim away from the light[29].

The change in the directionality of the droplets was investigated by manual z-tracking (Fig. 4b, c). We found that the droplets reorient in a direction characterized by the angle $\theta$ that defines a cone of probability for the reorientation (Fig. 4d). Larger values of the number of spiral turns N, measured at the moment of chiral inversion, are associated with larger reorientation angles $\theta$ (Fig. 4e). The precession angle between the path of the helical trajectory and the droplet axis also increases with N, and this affects both the radius R and pitch P of the helical trajectory during steady motion (as seen from Fig. 2f and Supplementary Fig. 11). Hence, we argue that the increase of precession angle with $N$ (Fig. 2a) is also key to define the reorientation angle, when the angular velocity reverts instantaneously. This understanding is in line with the key role played by precession angles in the swimming behavior of unicellular organisms[54–56].

We have investigated the behavior of the spiral droplets when helix inversion occurs slowly, i.e., when a lower intensity of irradiation is used (ca. 8 s). In such experimental conditions, the liquid crystal is allowed to transition in the compensated nematic state, where chirality is not expressed because the chiral contributions of the motor isomers compensate each other. As the number of spiral turns decreases in the droplet, the radius of the helical motion decreases and eventually the droplet swims along a straight trajectory that merges with the axis of the helical trajectory,

as shown in Fig. 4 where the achiral organization is associated with straight, linear motion. The angle between the helical and the straight trajectory was consistently smaller than the angle of reorientation that is associated with a sharp helix inversion (Fig. 5, Supplementary Movie 4). As the chiral information is gradually lost, there is enough time for the surfactant to redistribute around the droplet in the new configuration that appears gradually, therefore there is no flow induced by the diffusion of surfactant at the droplet surface associated with the reorganization of the spiral structure of the droplets, which would drive the droplet away from the initial direction of the helical motion.

## Discussion

The helical motion of spiral droplets in surfactant-rich water is determined by their handedness (which defines the handedness of the helical motion) and by the confinement of the liquid crystal helix in the droplets (which defines the pitch and radius of the trajectories). The system features transmission of chirality across length scales: the chirality of the molecular motors is transmitted to the spiral organization of the liquid crystal in the droplets, which is in turn transmitted into a chiral trajectory. Our experiments demonstrate that when the chiral dopants used to make the droplets are light-driven molecular motors, the light-induced inversion of handedness operated by the motor is associated with a deterministic reorientation of the droplets. Key to the emergence of directionality in response to light is the possibility of a dynamic conversion between right-handed and left-handed trajectories. The chirality-derived and deterministic reorientation of the spiral droplets shows similarities with the motile behavior of unicellular organisms[25] such as bacteria and the malaria parasite[57], as well as chiral neuronal motility[58]. In a context where accumulating evidence suggests that cells are intrinsically chiral and that the cytoskeleton and motor proteins steer their chiral motility[59], our findings show how asymmetric molecular operations are able to effectively regulate motility in fully artificial molecular systems.

## Methods

**Materials**. The nematic liquid crystal 5CB and the chiral dopant CB15 were purchased from Merck and used as received. TTAB (tetradecyl-trimethylammonium bromide) was purchased from Wako. The light-responsive chiral dopants Ph-**m** and Me-**m** were synthesized by following a procedure on which we reported earlier[22].

**Light-responsive chiral liquid crystals**. The chiral dopants (CB15, Me-**m**, or Ph-**m**) were weighed in glass vials and then the desired volume of nematic liquid crystal 5CB was added. The contents of the vial were heated above the iso-tropization temperature, where the opaque material becomes transparent. The isotropic mixture was stirred at 40°C and was cooled down to room temperature.

The light-responsive chiral liquid crystal prepared by this procedure was then introduced in a glass wedge cell (E.H.C. Co.) and its pitch was measured using the Grandjean Cano method, with a polarized optical microscope (Olympus BX51). The pitch of the liquid crystal helices doped with Me-**m** and Ph-**m** were measured in the dark, and subsequently at the photostationary state, by using $\lambda = 365$ nm collimated light from an LED (Thorlabs), until no change in texture was observed anymore (see Supplementary Fig. 2).

**Production of spiral droplets**. Motor-doped liquid crystals were injected with a microinjector (Femtojet, Eppendorf), using commercially available needles (Femtotips, Eppendorf) in an aqueous (25 wt % $D_2O$) surfactant solution (13 wt %, TTAB). The chamber containing the aqueous solution was prepared by applying an adhesive spacer ($\approx 1$ mm in thickness) onto a microscope slide. After injection, the chamber was closed with a glass coverslip. The size of the droplets was tuned by changing the injection pressure and injection time of the Femtojet.

**Tracking of the spiral droplets**. The droplets were imaged with an inverted bright field microscope (DMI6000B, Leica). The time-lapse images were acquired using a CMOS camera (HXG40, Baumer, 10 fps). Tracking was performed by adjusting the focus on the droplet manually. Experiments in which the droplets were out of focus were not included in the statistical analysis. The plane coordinates of the stage

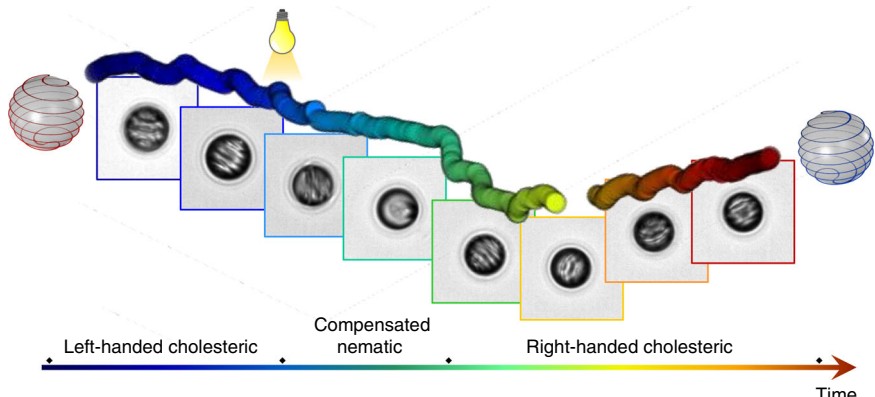

**Fig. 5** Influence of kinetics of chirality inversion on the reorientation. The spiral droplet shown here is doped with Me-**m**. When the light power is sufficiently low, the helical trajectory unwinds slowly and then rewinds with helix inversion. The insets show the structure of the droplet during motion, with straight motion being associated with the disappearance of the spiral organization in the compensated state. The direction of the motion is not modified significantly upon transition from helical to straight

(TANGO Desktop, Märzhäuser Wetzlar) and the Z coordinate of the motorized objective lens (× 20, Leica) were recorded using Labview. The spiral droplets were allowed to move over at least twice the pitch before irradiation started. Confocal fluorescence imaging was performed with a using a Nikon confocal microscope and laser sources with $\lambda = 488$ nm. The emission was recorded using appropriate dichroic mirrors and filter sets.

**Irradiation conditions**. The droplets were irradiated using high intensity (300 mW cm$^{-2}$) or low intensity (50 mW cm$^{-2}$) non-polarized collimated UV light ($\lambda = 365$ nm). Light intensity was measured using a PM-100D power meter (Thorlabs).

**Movie analysis**. The movies were analyzed using a custom-written program in python[38].

## Data availability
The authors declare that the data supporting the findings of this study are available within the paper and its supplementary information or from the corresponding authors upon reasonable request.

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

## Acknowledgements

N.K. acknowledges funding from the European Research Council (ERC Consolidator Grant 772564) and the Volkswagen Foundation (Integration of Molecular Components in Functional Macroscopic Systems 93424). Tk.Y. acknowledges funding from Grant-in-Aid for the JSPS Fellows (Grant No. 18J01239). M.S. acknowledges funding from JSPS KAKENHI Grant Number JP25103004, 16H02212.

## Author contributions

N.K. and M.S. supervised the research. F.L. performed the experiments. Tk.Y. built the tracking set-up. F.L. and Tk.Y analyzed the data. F.L. and A.R. prepared and characterized the light-responsive cholesteric liquid crystals. Td.Y. synthesized the molecular motors. N.K. and F.L. wrote the manuscript. All authors discussed the data, the interpretation, and the manuscript at all stages.

## Competing interests

The authors declare no competing interests.
