## [Peer Review File · Nature Communications]

Reviewers' comments:

Reviewer #1 (Remarks to the Author):

In this paper the authors primarily report on switchable helical swimming in liquid crystal droplet swimmers with photoactive chiral dopants.

They have previously published similar, non-switchable chiral swimming using water insoluble chiral dopants, the novelty in this study involves a) the water soluble dopant CB15 which causes a different chirality evolution b) using light switchable dopants to reorient the droplets at will, claiming deterministic reorientation angles.

The idea is neat, but the manuscript is not organised well enough to convincingly support these rather bold statements and, frankly, does not appear sufficiently finished for a high quality journal submission. I think there's too much to be cleaned up for a second round of reviews and I'd recommend a fresh start on the paper with more clear statistics.

A non-exhaustive list of the issues I found:

-I have trouble in judging the quality of the experimental statistics. There are multiple claims of 'typical behaviour', but the supporting plots either show single droplet data or do not state over how many droplets these are ensemble averaged. Information necessary for error estimation like polydispersity or dopant concentration is missing. 3D data acquisition is apparently done by manual z tracking, which is highly dependent on the skill of the experimentalist. The helices documented in the supporting information look rather nonuniform, which clashes with the impressively low error margins in the MS. Data selection or 'cherrypicking' is permissible, especially with variable experimental conditions, but needs to be justified. With respect to the SI plots, the tick marks are illegible, and the positions of boundaries should be marked (i.e., do some of these droplets get trapped at the walls?)

-The statement that helical swimming is suppressed for $N < 14$ refers to Fig. 2d), but all I can see there is a divisor line with the words 'straight' and 'helical', which does not constitute evidence. It is also again not apparent whether this is ensemble averaged, and it would be much more illustrative to plot this over radius than time.

As mentioned panel 2e) shows regular helical swimming ('period' is a time, btw, I assume the authors mean 'pitch'), but I fail to see any signature of that in the SI's example trajectories, which appear to be for different droplet types in any case.

Similarly, I'd take the 'deterministic' reorientation in Fig. 3 with a grain of salt. Why is the distribution in panel d exactly symmetric around $\theta = 0$?

I don't understand the frustratingly vague explanation for the reorientation mechanism in page 9. E.g., why would a higher torque lead to a larger radius? I'd expect the opposite. What do they mean with 'more confined' in terms of N ?

-I would not call these droplets protocells, they lack membranes or internal functionalisation

Reviewer #2 (Remarks to the Author):

This manuscript is centered on the behaviour of droplets of a nematic liquid crystal doped with a chiral molecular motor, in water solutions that contain a surfactant. It is shown that the droplets move along a helical trajectory, whose handedness can be controlled by light, exploiting the change in helicity of the chiral dopants under photoisomerization.

This study provides new insights on a topic of high current interest for a wide readership. The paper has an acceptable form, the figures are clear and adequately described. I think that this paper can be published in Nature Commun. after some revision. Some points that in my opinion could be considered are reported below.

1) I have concerns about the use of some words and expressions:

- What is the reason for using the word 'protocells'? Shouldn't one simply speak of droplets?

- The use of the word 'helix' throughout the paper can lead to some confusion, especially in readers who are not familiar with liquid crystals.

I refer in particular to expressions as 'helix-based dynamic liquid crystals' (page 3) and 'droplets of liquid crystalline helices' (caption to Fig. 2), which could be misleading, since readers could have the impression that droplets contain helical molecules. But this is not the case here (unless motor dopants are described as 'helices'). Therefore I suggest to rephrase such expressions.

- The meaning of the 'shape persistent' (see for instance 'CB15 is a shape-persistent chiral molecule') is not fully clear. Isn't it sufficient to say 'chiral'? By the way, CB15 is a flexible molecule, therefore it can change its shape, although it does not change its absolute configuration.

2) In the manuscript one finds expressions such as: 'spiral shape of the droplet' and 'spiral droplet'. This gives the impression that droplets have a helical shape and that this is important for the phenomenon under investigation. But from the figures droplets seem to have a spherical shape. I wonder if the authors rather mean the internal organization of the liquid crystal director (which is different from the droplet shape).

This issue (spiral shape of droplets or chiral organization of the liquid crystal within the droplets) needs to be clearly addressed.

4) The mechanism of the phenomenon under investigation is not obvious and the explanations are not fully clear to me. For instance, what is the 'chiral flow inside the droplet'? What is flowing: liquid crystal molecules, dopants? Is this flow crucial for the spiral motions?

Reviewer #3 (Remarks to the Author):

The authors demonstrate an interesting LC droplet based protocell that is able to change the direction of its helical translational motion via optical switching of its chiral structure. The experimental results are clearly described, unfortunately the mechanisms that enable translational motion of protocells along helices and their redirection induced by optical switching of chiral structures of their molecular motors are poorly explained. I have the impression that the authors are trying to attract the readers by presenting interesting behavior of their protocells but forgetting that providing the understanding of the driving process is crucial for a paper published in a journal like Nature Communications. Below I list more details and some additional points that need attention. The paper should be improved before I can recommend its publication in Nature Communications.

Page 3

The claim "We started by incorporating light-driven molecular motors in droplets of helix-based dynamic liquid crystals (Figure 1)." does not correspond to the description in the Methods. The object "helix-based dynamic liquid crystal" is also not explained!

Although a lot is known about chiral nematic structures in droplets the structure "spiral droplet" that seems to be essential for observed phenomena needs to be clearly described in text and figure. As structures and their stability-range strongly depends on the surface anchoring it should

be specified

particularly in relation to the surfactant that is not homogeneously distributed.

Figure 1 is poorly explained! It needs better descriptions of all presented segments.

- m and m^* are practically hardly recognized.
- The modeled pictures need detailed explanations. The red and blue lines are even not mentioned (in case of strong homeotropic anchoring there are disclination lines) etc.
- The color-coding of the spiral-like trajectories is missing.

Pages 3-4

The descriptions of the of driving mechanisms on Pages 3

“When the chirality of the droplet is coupled to gradients in interfacial tension, a chiral flow propels these chiral entities forward in a screw-like movement.³² In this system, light activates a cascade of bottom-up events, in which the helical aspect of the molecular motors is converted into the helical organization of a chiral liquid crystal, amplified into the spiral shape of the droplet, and further into a chiral flow of fluid matter that re-orientates the movement in a deterministic manner, upon activation of the molecular motor.”

and 4

“Key to the establishment of gradients in interfacial tension is the presence of a large quantity of surfactants in water, forming micelles that can solubilize small amounts of liquid crystals and thus create inhomogeneities in surface tension that are compensated by a flow driving the droplets forward, a phenomenon known as the Marangoni effect.³⁶ The mechanism can be understood as follows: the solubilization of the liquid crystal as mediated by the surfactant creates an instability, and the gradient in surface tension that is associated with this instability initiates a flow around the droplet, that effectively propels the droplet forward. The spiral shape of these droplets mediates a non-equilibrium cross-coupling between the gradient of interfacial tension and a chiral flow inside the droplet, that results in propagation following a helical path (Figure 2a,b, Supplementary Video 1).”

should be improved! Including origin of Marangoni flows, induction of droplet rotation, propelling of droplets, spiral trajectory, and orientation of droplets, rotation axis & velocity.

Using the term “spiral shape” of droplets is confusing as only director structures exhibit spiral like shapes.

Page 4

The description “helical organization of the molecules in three dimensions” is not a good explanation for a simple bulk cholesteric with winding in one direction.

Is the motion essentially 2D as drops move several mm in a millimeter thick layer? How spiral axis of the motion is kept in plane? If not what is the effect of collision to the surfaces? Are the drops tracked in 3D?

Page 5

Fig. 2: Is the droplet structure successfully tracked along the trajectory?

Page 6

The symbols for the radius and period of the trajectory should be introduced in the text!

Shrinking of drops and increasing concentration of motors needs explanation. Do only 5CB and 15CB dissolve in water forming micelles? What happens to molecular motors?

Page 7

Why larger droplets move faster (Figure 2g) should be commented.

Describing strong irradiation with light (ca. 1 s) should be more precise: specifying wavelength, polarization, and intensity.

How fast the direction of the translation can change? How fast is the transition of the droplet structure? How different is the pitch after the inversion? Certainly N changes as droplets shrink with time.

What is deciding how a droplet restructure (flow, light intensity...)? Is the new structure always spiral? Are there also intermediate structures?

Pages 8, 9 &10

How the motion is confine to a thin layer? Is there a mechanism keeping redirection mostly in 2D? Effect of boundaries?

The low intensity light transition goes via chiral states where N is below the threshold for the appearance of spiral structures. It is no need to be called nematic in Fig. 4

Reviewer #1 (Remarks to the Author):

In this paper the authors primarily report on switchable helical swimming in liquid crystal droplet swimmers with photoactive chiral dopants.

They have previously published similar, non-switchable chiral swimming using water insoluble chiral dopants, the novelty in this study involves a) the water soluble dopant CB15 which causes a different chirality evolution b) using light switchable dopants to reorient the droplets at will, claiming deterministic reorientation angles.

The idea is neat, but the manuscript is not organised well enough to convincingly support these rather bold statements and, frankly, does not appear sufficiently finished for a high quality journal submission. I think there's too much to be cleaned up for a second round of reviews and I'd recommend a fresh start on the paper with more clear statistics.

***Our response to this overall comment:** We accept the feedback that the presentation of the arguments needs to be much clearer and better organized. We have consequently performed an extensive re-draft of the manuscript, and trust R#1 will find the data convincing and its presentation improved.*

It may be worth noting that we are not studying living organisms (e.g. E coli or other unicellular swimmers), therefore statistical treatment is not our primary tool for interpreting behavior. Rather, for our experiments, we find that optical microscopy can provide insightful information on our wholly artificial chemical systems, while concurring with R#1 that semi and fully automated image analysis will be useful to investigate these processes further. In light of the feedback by R#1, we recognize that some of our data presentation lacked clarity, and have acted on the suggestions for improvement.

Below we address the specific comments made by R#1. We have also included more statistics, completed the Supporting Information, and further revised the manuscript, which is now substantially improved. Based on this, we consider that the paper is now worthy of publication, as indicated in our cover letter to the Editor.

A non-exhaustive list of the issues I found:

-I have trouble in judging the quality of the experimental statistics. There are multiple claims of 'typical behaviour', but the supporting plots either show single droplet data or do not state over how many droplets these are ensemble averaged. Information necessary for error estimation like polydispersity or dopant concentration is missing.

***Response:** We accept these comments by R#1, and we now provide error bars (standard deviation) for the time evolution of the chiral confinement N (Figure 2e).*

We also provide specific information on size-dispersity in Supplementary Figure 1 (see below for Supplementary Figure 1).

*Furthermore, the dopant concentration and the error estimation are now specified in the main text (Page 7), while in our first submission this information was included in the supplementary information. Specifically, the concentrations are $(10.00 \pm 0.01)\text{wt}\%$, $(0.38 \pm 0.01)\text{wt}\%$ and $(0.68 \pm 0.01)\text{wt}\%$ for CB15, Me-**m** and Ph-**m**, respectively.*

Supplementary Figure 1. Polydispersity of the cholesteric droplets produced by injection in the aqueous medium.

We also accept that the word “typical” is confusing because it is unclear whether it refers to a single behavior, or to an average. We have removed this word from the manuscript, and specified in each caption whether we show individual or averaged behavior.

3D data acquisition is apparently done by manual z tracking, which is highly dependent on the skill of the experimentalist. The helices documented in the supporting information look rather nonuniform, which clashes with the impressively low error margins in the MS. Data selection or 'cherry-picking' is permissible, especially with variable experimental conditions, but needs to be justified.

Response: *We have accommodated the request to show detailed data. In particular, we have modified Figure 2 extensively to also include individual trajectories (new Figure 2d). Also, R#1 is correct in the assumption that the drops get trapped when interacting with the boundaries of the chamber and this we did not specifically report in the previous version of the manuscript. We now clearly specify in the text that we excluded from the analysis those sections of the trajectory where the drops are trapped at the interface, because they are not representative of the movement that operates in the bulk of the chamber.*

We write: “When the droplets reached the top or bottom boundaries of the chamber, they became trapped and consequently started swimming in circles, similarly to what living microorganisms do (for a case involving Ph-m droplets see Figure 2d right panel). In this manuscript, trapped trajectories and other sections of 2D trajectories, where droplets interact with the boundaries of the chamber, have been excluded from the statistical analysis.”

In the caption of Figure 2 we specify: “The data acquisition is done by manual z-tracking”.

We appreciate that $R \neq 1$ refers to our standard deviation as ‘impressively low’, however these deviations are arguably not as low as if automated.

We also note that the trajectories we showed in the Supporting Information were not representative of the droplets doped with CB15 – we apologize that this was not clear. We have now added all the trajectories corresponding to the CB15 droplets (Supplementary Figure 4). What minimizes experimental deviations in the case of these CB15-doped drops is the fact that the solubilization of the cholesteric liquid crystal occurs together with that of the dopant CB15. This ‘homogeneous’ solubilization process allows for a high reproducibility in the geometry and kinetics of the helical motion and therefore, during experiments, the geometrical parameters associated with the propulsion of these droplets were always accessible to the operator.

With respect to the SI plots, the tick marks are illegible, and the positions of boundaries should be marked

***Response:** We now specify in the figure and/or figure captions of the Supporting Information what is the size of the grid for the XYZ coordinate system (i.e. $200\ \mu\text{m} \times 200\ \mu\text{m} \times 200\ \mu\text{m}$).*

-The statement that helical swimming is suppressed for $N < 14$ refers to Fig. 2d), but all I can see there is a divisor line with the words 'straight' and 'helical', which does not constitute evidence. It is also again not apparent whether this is ensemble averaged, and it would be much more illustrative to plot this over radius than time.

***Response:** This point concerns Figure 2e in the re-draft. We now specify that the data is ensemble averaged in the caption. Also, we have added three individual trajectories in Figure 2, and we have added all individual trajectories for CB15-doped droplets in the Supporting Information (Supplementary Figure 4), which amounts to a total of six individual traces.*

*Furthermore, we accommodate the feedback of Rev#1 by rewriting the paragraph “Helical propulsion of swimming droplets” to improve its clarity and to offer a detailed discussion on the transition between chiral and non-chiral motion for both Me-**m** and CB15.*

From our revised manuscript: “Cholesteric liquid crystals doped with (10.00 ± 0.01) wt% CB15 had an initial pitch of ca. $1.5\ \mu\text{m}$, with initial values of N ca. 20, and with such a confinement ratio the droplets start swimming helically as soon as they are injected in solution (Figure 2d). During movement, the number of spiral disclination lines decreases until eventually the droplet becomes nematic and swims in a straight fashion (Figure 2d, left panel). The decrease of the number of spiral lines indicates that the pitch of the cholesteric helix increases over time, which we attribute to the co-solubilization of CB15 with its liquid crystal host, by the empty micelles in water (Supplementary Figure 2). In contrast, cholesteric droplets doped with (0.38 ± 0.01) wt% Me-**m** have an initial pitch of $4.3\ \mu\text{m}$ ($N = 10$), and over the duration of the motion the Me-**m** concentrates in the droplet, as evidenced by the increase in the number of spiral turns.”

And: “The helical twisting power of Ph-**m** is larger than that of Me-**m** in the same liquid crystal, and therefore Ph-**m** was used as a dopant to form droplets of initial pitch ca. $1.6\ \mu\text{m}$ ($N = 25$, 0.68 ± 0.01 wt% of Ph-**m**). These droplets are characterized by a similar confinement ratio than the CB15-doped drops, and therefore they also immediately start along helical trajectories. However, instead of decreasing, the chiral confinement increases over time in these droplets, until the high concentration of motor disrupts the liquid crystal order and the droplet consequently becomes isotropic (Figure 2d, right panel).”

We have added the time evolution of the droplet structure in Supplementary Figure 3 to show and support the different behavior of the droplets doped with CB15, Me-**m**, and Ph-**m**. (please see below for the figure). We also provide a plot of N over radius in Figure 2f.

Supplementary Figure 3. Evolution of the droplet in time, for different chiral dopants. *a*, Droplets doped with CB15 initially display a well-defined spiral distribution of the director field. In time, the number of spiral disclination lines decreases and eventually (after ~ 500 s), the chirality of the droplet is not expressed at the structural level anymore, i.e. the droplet can be considered pseudo-nematic. *b*, Droplets doped with Me-m need time to organize into the spiral configuration, after what they start swimming along helical trajectories (here after ~ 300 s) and eventually the droplets become isotropic. *c*, Droplets doped with Ph-m swim helically after being created and they maintain a clear helical trajectory until isotropization occurs at 337s. The occurrence of isotropization for Ph-m and Me-m doped cholesteric droplets suggests that the dopant is concentrating in the droplet over time, in contrast to what happens when CB15 is used as a dopant. Scales bar are $10 \mu\text{m}$.

As mentioned panel 2e) shows regular helical swimming ('period' is a time, btw, I assume the authors mean 'pitch'), but I fail to see any signature of that in the SI's example trajectories, which appear to be for different droplet types in any case.

Response: We have replaced “period” with “pitch” throughout the text, and we thank R#1 for pointing out to this mistake. Besides, we and have included all trajectories for CB15 doped droplets in the Supplementary Information (Supplementary Figure 4).

Similarly, I'd take the 'deterministic' reorientation in Fig. 3 with a grain of salt. Why is the distribution in panel d exactly symmetric around $\theta=0$?

Response: Indeed, we accept that the presentation choice we made can create confusion – R#1 is right that the angle θ carries meaning only between 0 and 180 degrees. We initially decided to mirror the data in order to point to the reader that there is a three-dimensional cone of probability that is associated with each value of angle θ . Primarily our intention was to not claim more than what we can, i.e. we did not want to suggest that the droplet has a specific direction of re-orientation in three dimensions. However, we agree with R#1 that the choice we made is confusing and we now only show the data as measured.

We have added in the caption of Figure 4d: “These values are meaningful only between 0 and 180, which means that a cone of deterministic reorientation is associated to each re-orientation angle theta.”.

I don't understand the frustratingly vague explanation for the reorientation mechanism in page 9. E.g., why would a higher torque lead to a larger radius? I'd expect the opposite. What do they mean with 'more confined' in terms of N?

Response: *We apologize for our explanation being perceived as “vague”, and in our re-draft we have endeavored to be as specific as we can, while providing a rationale that remains understandable for the readership of Nature Communications. However, we would like to stress that this publication remains a communication, and it reports an entirely new and unexpected phenomenon for artificial systems. We are currently developing a complete model to further explore these findings; however, we firmly believe that providing a detailed model and a quantitative explanation reaches beyond the purpose of this manuscript.*

In addition to this extensive re-write, and in order to further support our mechanistic explanation based on the chiral distribution of surfactants at the surface of the droplet, we have also included the confocal fluorescence image of a cholesteric droplets covered by fluorescently labeled surfactants (see Supplementary Figure 12).

Supplementary Figure 12. Confocal image of a cholesteric droplet doped with CB15 and with homeotropic anchoring conditions induced by TTAB and a fluorescently labelled surfactant BODIPY™ FL C16 (4,4-difluoro-5,7-dimethyl-4-bora-3a,4a-diaza-s-indacene-3-hexadecanoic acid). The fluorescently-labelled surfactant follows the chiral structure of the droplet surface, *i.e.* it is concentrated in the area between two disclination lines and is less concentrated in the disclination lines. Scale bar is 50 μm .

-I would not call these droplets protocells, they lack membranes or internal functionalization

Response: *We accept this comment on terminology, and have changed all mentions of protocells to droplets.*

Reviewer #2 (Remarks to the Author):

This manuscript is centered on the behaviour of droplets of a nematic liquid crystal doped with a chiral molecular motor, in water solutions that contain a surfactant. It is shown that the droplets move along a helical trajectory, whose handedness can be controlled by light, exploiting the change in helicity of the chiral dopants under photoisomerization.

This study provides new insights on a topic of high current interest for a wide readership. The paper has an acceptable form, the figures are clear and adequately described. I think that this paper can be published in Nature Commun. after some revision. Some points that in my opinion could be considered are reported below.

***Response to this overall comment:** We thank R#2 for the time spent on our manuscript and the positive evaluation of the paper.*

We value R#2's feedback on how we might further improve the clarity of the paper, in particular the comments that she/he made with regard to terminology. Below we address those points and how we have used them to improve the manuscript.

1) I have concerns about the use of some words and expressions: - What is the reason for using the word 'protocells'? Shouldn't one simply speak of droplets?

***Response:** We accept the comment (also made by another reviewer) and we have changed all mentions of 'protocells' to 'droplets' throughout the manuscript.*

- The use of the word 'helix' throughout the paper can lead to some confusion, especially in readers who are not familiar with liquid crystals. I refer in particular to expressions as 'helix-based dynamic liquid crystals' (page 3) and 'droplets of liquid crystalline helices' (caption to Fig. 2), which could be misleading, since readers could have the impression that droplets contain helical molecules. But this is not the case here (unless motor dopants are described as 'helices'). Therefore I suggest to rephrase such expressions.

***Response:** We recognize that our mixed use of terminology can result in misunderstanding. In our re-draft, we systematically use 'cholesteric liquid crystals' instead of 'helix based liquid crystals'.*

- The meaning of the 'shape persistent' (see for instance 'CB15 is a shape-persistent chiral molecule') is not fully clear. Isn't it sufficient to say 'chiral'? By the way, CB15 is a flexible molecule, therefore it can change its shape, although it does not change its absolute configuration.

***Response:** We welcome this feedback – indeed we meant that CB15 does not change configuration under illumination. As per suggestion of R#2, now CB15 is simply called 'chiral dopant'.*

2) In the manuscript one finds expressions such as: 'spiral shape of the droplet' and 'spiral droplet'. This gives the impression that droplets have a helical shape and that this is important for the phenomenon under investigation. But from the figures droplets seem to have a spherical shape.

I wonder if the authors rather mean the internal organization of the liquid crystal director (which is different from the droplet shape).

This issue (spiral shape of droplets or chiral organization of the liquid crystal within the droplets) needs to be clearly addressed.

***Response:** The surface corrugation of the droplet follows the director distribution, and therefore the droplet really has a spiral shape because the director field has a spiral organization. However, we agree with R#2 that the shape is probably not as important here as the internal spiral organization, because it is the internal spiral organization that gives the Marangoni flow a chiral twist. In the revised version of the manuscript, we removed any mention of the "shape" of the chiral droplet.*

4) The mechanism of the phenomenon under investigation is not obvious and the explanations are not fully clear to me. For instance, what is the 'chiral flow inside the droplet'? What is flowing: liquid crystal molecules, dopants? Is this flow crucial for the spiral motions?

Response: *We understand that the explanations we provide are not complete. Our primary concern was to not make claims that we cannot prove or not yet support with models – as this phenomenon is new and unexpected, more research will come to elucidate its mechanism quantitatively. However, we have re-written the paragraph in which we provide a rationale for the mechanism (see the redrafted manuscript). We are, of course, more than happy to clarify the text further if needed and we welcome any suggestion in this regard.*

Further, the comments from R#2 highlight the need for us to be very clear on our use of terminology. Indeed, yes, the chiral flow of the liquid crystal (= liquid crystal molecules and dopants together), is essential to impart angular velocity. The coupling between the angular velocity and the translational motion (both induced by Marangoni flow) results in the helical motion. So, in our understanding, this flow is in fact essential to the establishment of helical motion.

Reviewer #3 (Remarks to the Author):

The authors demonstrate an interesting LC droplet based protocell that is able to change the direction of its helical translational motion via optical switching of its chiral structure. The experimental results are clearly described, unfortunately the mechanisms that enable translational motion of protocells along helices and their redirection induced by optical switching of chiral structures of their molecular motors are poorly explained. I have the impression that the authors are trying to attract the readers by presenting interesting behavior of their protocells but forgetting that providing the understanding of the driving process is crucial for a paper published in a journal like Nature Communications. Below I list more details and some additional points that need attention. The paper should be improved before I can recommend its publication in Nature Communications.

***Response to this overall comment:** We accept the feedback that our explanations can be improved, also in line with the comments by other reviewers. We have used the feedback and commentary listed below for taking restorative action, as described. As a consequence, the manuscript has now been substantially improved and we trust that R#3 will now find that its contents are at the right level of detail for the readership of Nature Communications.*

Page 3 The claim “We started by incorporating light-driven molecular motors in droplets of helix-based dynamic liquid crystals (Figure 1).” does not correspond to the description in the Methods. The object “helix-based dynamic liquid crystal” is also not explained!

***Response:** We accept how this terminology can lead to misunderstanding, particularly to readers who are familiar with liquid crystals. We have amended Figure 1 to show that it is the doping of a nematic liquid crystal with molecular motors that yields a light-responsive cholesteric liquid crystal. To avoid further confusion, we also have replaced “helix-based dynamic liquid crystals” with “light-responsive cholesteric liquid crystals”*

The description of Figure 1 now reads: “We started by incorporating light-driven molecular motors in a nematic liquid crystal to induce a light-responsive cholesteric liquid crystal, that was confined in spherical droplets (Figure 1)”.

Although a lot is known about chiral nematic structures in droplets the structure “spiral droplet” that seems to be essential for observed phenomena needs to be clearly described in text and figure. As structures and their stability-range strongly depends on the surface anchoring it should be specified particularly in relation to the surfactant that is not homogeneously distributed.

***Response:** We agree with R#3 that the anchoring of the liquid crystals at the droplet/air interface is key to droplet morphology and we mention this in the text specifically. We now write: “This light-responsive material was confined into a spherical droplet with perpendicular anchoring – meaning that the liquid crystal molecules are oriented perpendicular to the interface”.*

We have added references #35 and #36 on the organization of cholesteric liquid crystals in droplets, with a detailed description of the director distribution and of its stability, depending on the anchoring conditions:

Krakhalev, M. N. et al. Orientational structures in cholesteric droplets with homeotropic surface anchoring. *Soft Matter*, **15**, 5554-5561 (2019).

Seč, D., Porenta, T., Ravnik, M. & Žumer, S. Geometrical frustration of chiral ordering in cholesteric droplets. *Soft Matter*, **8**, 11982-11988 (2012).

In addition, we note that two other references (#33 and #34 in the re-draft), that were already included in the original manuscript and can also offer insight on the same subject:

Seč, D., Čopar, S. & Žumer, S. Topological zoo of free-standing knots in confined chiral nematic fluids. Nat. Commun. 5, 3057 (2014).

Orlova, T., Abhoff, J. S., Yamaguchi, T., Katsonis, N., Brasselet, E. Creation and manipulation of topological states in chiral nematic microspheres. Nat. Commun. 6, 7603 (2015).

Figure 1 is poorly explained! It needs better descriptions of all presented segments.

- m and m^* are practically hardly recognized.
- The modeled pictures need detailed explanations. The red and blue lines are even not mentioned (in case of strong homeotropic anchoring there are disclination lines) etc.
- The color-coding of the spiral-like trajectories is missing.

Response: We agree, and have modified Figure 1 to make it clearer and its contents are also explained in more detail both in the text and in the caption.

Motor m and its metastable form m^* are now noted with larger characters. The arrow is specified to be the time evolution of the trajectory as indicated by color coding in Figure 1.

We also explain that the lines at the surface of the droplet are disclination lines: “This light-responsive material was confined into a spherical droplet with perpendicular anchoring – meaning that the liquid crystal molecules are oriented perpendicularly to the interface. In these conditions of interface anchoring, a double spiral disclination line forms at the interface, that is associated with a spiral distribution of the director field in the bulk.^{i,ii,iii,iv} The handedness of this double spiral reflects the handedness of the cholesteric phase.”

New caption: “**Figure 1. Design of chiral droplets that respond to light with a deterministic reorientation of their swimming motion.** The chirality of molecular motor m and that of its light-induced metastable form m^* , are amplified across increasing length-scales in liquid crystals to yield cholesteric helices (left- handed in red, right- handed in blue). In spherical confinement and with the liquid crystal molecules aligning perpendicularly to the interface, a double spiral disclination line appears at the surface of the droplet. This spiral disclination line is visible under polarized optical microscopy (scale bar 50 μm) and is also shown in the model of the droplet (red for left-handed and blue for right-handed). These chiral droplets are propelled along helical trajectories with a handedness that is opposite that of the droplets. Upon illumination, the molecular motors invert their helicity, in a process that drives their rotation and is associated with inversion of their helical trajectory and with a directional re-orientation.”

Pages 3-4

The descriptions of the of driving mechanisms on Pages 3 “When the chirality of the droplet is coupled to gradients in interfacial tension, a chiral flow propels these chiral entities forward in a screw-like movement.³² In this system, light activates a cascade of bottom-up events, in which the helical aspect of the molecular motors is converted into the helical organization of a chiral liquid crystal, amplified into the spiral shape of the droplet, and further into a chiral flow of fluid matter that re-orientes the movement in a deterministic manner, upon activation of the molecular motor.”

and 4 “Key to the establishment of gradients in interfacial tension is the presence of a large quantity of surfactants in water, forming micelles that can solubilize small amounts of liquid crystals and thus create inhomogeneities in surface tension that are compensated by a flow driving the droplets forward, a phenomenon known as the Marangoni effect.³⁶ The mechanism can be understood as follows: the solubilization of the liquid crystal as mediated by the surfactant creates an instability, and the gradient in surface tension that is associated with this instability initiates a flow around the droplet, that effectively propels the droplet forward. The spiral shape of these droplets mediates a non-equilibrium cross-coupling between the gradient of interfacial tension and a chiral flow inside the droplet, that results in propagation following a helical path (Figure 2a,b, Supplementary Video 1).”

should be improved! Including origin of Marangoni flows, induction of droplet rotation, propelling of droplets, spiral trajectory, and orientation of droplets, rotation axis & velocity.

***Response:** After it being pointed out to us by R#3, we recognize and accept that the explanations in the submitted manuscript were potentially confusing. We have rewritten the text extensively (see changes in the manuscript).*

Using the term “spiral shape” of droplets is confusing as only director structures exhibit spiral like shapes.

***Response:** We agree with R#3 and have addressed this point by using the expression “spiral droplets”, without any specific reference to the shape of these drops.*

We write: “Here, the control of molecular chirality by light activates a cascade of bottom-up events, because the axial chirality of the molecular motors is amplified into the cholesteric helix, further into the spiral organization of the director field in the droplet, and eventually into a chiral flow of fluid matter that re-orientes the movement upon light-induced inversion of the axial chirality of the molecular motors.”

And later: “Here, the spiral organization of the molecules in the droplets accounts for rotational flow, which gives the droplet an angular velocity.”

Page 4. The description “helical organization of the molecules in three dimensions” is not a good explanation for a simple bulk cholesteric with winding in one direction.

***Response:** We accept this comment and have addressed this point.*

We write: “The chirality of the motors induces a twist in the liquid crystal organization and leads to the formation of a light-responsive cholesteric liquid crystal, that is characterized by a helical organization of the molecules (Figure 1).”

Is the motion essentially 2D as drops move several mm in a millimeter-thick layer?

***Response:** The 20 μm diameter droplets are confined in a chamber which is 1 mm thick and 5 mm in diameter. The differences in size are such that the motion is three dimensional.*

How spiral axis of the motion is kept in plane? If not what is the effect of collision to the surfaces?

Response: *The drops are free to move in 3D and therefore the reviewer is right that they can collide against the top or bottom part of the chamber (which are glass slides). When that happens, they are trapped and move along circular trajectories. We have described this situation in the revised manuscript (Figure 2) and we have specified in the text that these interfacial trapping events were not considered in our statistics.*

Are the drops tracked in 3D?

Response: *In the methods section we now specify: “During motion, 3D tracking was achieved by adjusting the focus on the droplet manually.”*

Page 5, Fig. 2: Is the droplet structure successfully tracked along the trajectory?

Response: *Yes, we confirm that the droplet is successfully tracked along the trajectory. We added: ‘Experiments where the droplets appeared out of focus were discarded.’*

Page 6. The symbols for the radius and period of the trajectory should be introduced in the text!

Response: *The symbols for the radius and period are now included in the text, as follows: “The coupling between the flow induced by the instability and the chiral flow in the droplet results in a helical trajectory that is characterized by a pitch P and radius R (Figure 2a,b and Supplementary Video 1).”*

Shrinking of drops and increasing concentration of motors needs explanation. Do only 5CB and 15CB dissolve in water forming micelles? What happens to molecular motors?

Response: *Our results indicate that Me-**m** and Ph-**m** motors are not taken up during the solubilization operated by the surfactant.*

*We now write: “In contrast, for cholesteric droplets doped with (0.38 ± 0.01) wt% Me-**m** and having an initial pitch of $4.3 \mu\text{m}$ ($N = 10$), Me-**m** concentrates in the droplet during the motion and translates into an increase of spiral turns. This observation is consistent with previous report on chiral dopants structurally more complex than CB15, which have been proven to enrich in shrinking droplets.^{Error!} Bookmark not defined.”*

Page 7. Why larger droplets move faster (Figure 2g) should be commented.

Response: *We now comment in depth on the size-dependent velocity of the droplet.*

We write: “The observation that larger droplets swim faster agrees with trends that were established for isotropic oil drops in water (Figure 3b).^v This relation between velocity and size is related to the asymmetry in the surroundings of a droplet: for the propulsion driven by the Marangoni effect, a front-rear asymmetric distribution of surfactant at the surface of the droplet is required.^{vi} When the size of the droplet is small, the supply of surfactant molecules from the surrounding solution can readily reduce such an inhomogeneous distribution. In contrast, for larger droplets, the supply is less sufficient to reduce the inhomogeneity. Therefore, the speed of the droplet decreases with decreasing droplet size.”

Describing strong irradiation with light (ca. 1 s) should be more precise: specifying wavelength, polarization, and intensity.

Response: *We apologize for the confusion - the irradiation is in fact continuous, and the ca. 1s timescale indicates the time that is need for the droplet to invert its chirality. We now have a separate paragraph to describe the irradiation conditions: wavelength and intensity are specified in the methods section.*

Also, we write: “Under illumination, the operation of the molecular motors inverts the cholesteric helix in the droplets and induces the same absolute values of pitch. Remarkably, when the handedness of the spiral is forced to invert rapidly, by strong irradiation with light (ca. 1 s for the overall inversion), the droplets re-orient significantly in three-dimensional space [...].”

How fast the direction of the translation can change? How fast is the transition of the droplet structure? How different is the pitch after the inversion? Certainly N changes as droplets shrink with time.

Response: *The droplet changes direction as soon as the chirality is fully inverted (i.e. ca. 1 s). The pitch has opposite sign than the starting pitch and same absolute value. Within the timescale of the inversion N does not change considerably.*

We write: “We note that helix inversion occurs in one second and moreover, the final pitch is equal to the initial pitch, therefore the confinement ratio N does not vary considerably within the timescale of the inversion. Specifically, and considering the time evolution of N (see Figure 2e), we estimate that N increases of only 0.03 units in one second.”

What is deciding how a droplet restructure (flow, light intensity...)? Is the new structure always spiral? Are there also intermediate structures?

Response: *The spiral director field of the droplet reconfigures depending on the illumination conditions. Since we use strong irradiation the photostationary state is reached rapidly and therefore the new cholesteric helix has the same pitch than before irradiation, but with opposite handedness. As the transition from one chiral state to the other, even though extremely fast, is not immediate, the droplet must at a certain point pass through a pseudo-nematic state after unwinding of the cholesteric helix and before rewinding the cholesteric with different handedness. However, the timescale of the overall inversion (ca. 1 s) make the intermediate structures irrelevant for the propulsion. Notably, the pseudo-nematic state is observed when lower irradiation intensities are used (see Figure 5).*

Pages 8, 9 & 10. How the motion is confined to a thin layer? Is there a mechanism keeping redirection mostly in 2D? Effect of boundaries?

Response: *R#3 is correct in the assumption that the drops get trapped when interacting with the boundaries of the chamber and this we did not specifically report in the previous version of the manuscript. Essentially, we have modified Figure 2 extensively to include individual trajectories (new Figure 2d). Also, we now clearly specify in the text that we excluded from the analysis those sections of the trajectory where the drops are trapped at the interface, because they are not representative of the movement that operates in the bulk of the chamber.*

We write: “When the droplets reached the top or bottom boundaries of the chamber, they became trapped and consequently started swimming in circles, similarly to what living microorganisms do (for a case involving Ph-m droplets see Figure 3d right panel). In this manuscript, trapped trajectories and other sections of 2D trajectories, where droplets interact with the boundaries of the chamber, have been excluded from the statistical analysis.”

In the caption of Figure 2 we specify: “The data acquisition is done by manual z-tracking”.

The low intensity light transition goes via chiral states where N is below the threshold for the appearance of spiral structures. It is no need to be called nematic in Fig. 4.

Response: *We modified “nematic” into “pseudo-nematic”.*

-
- ⁱ Seč, D., Čopar, S. & Žumer, S. Topological zoo of free-standing knots in confined chiral nematic fluids. *Nat. Commun.* **5**, 3057 (2014).
- ⁱⁱ Orlova, T., Abhoff, J. S., Yamaguchi, T., Katsonis, N., Brasselet, E. Creation and manipulation of topological states in chiral nematic microspheres. *Nat. Commun.* **6**, 7603 (2015).
- ⁱⁱⁱ Krakhalev, M. N. *et al.* Orientational structures in cholesteric droplets with homeotropic surface anchoring. *Soft Matter*, **15**, 5554-5561 (2019).
- ^{iv} Seč, D., Porenta, T., Ravnik, M. & Žumer, S. Geometrical frustration of chiral ordering in cholesteric droplets. *Soft Matter*, **8**, 11982-11988 (2012).
- ^v Ueno, N. *et al.* Self-propelled motion of monodisperse underwater oil droplets formed by a microfluidic device. *Langmuir* **33**, 5393-5397 (2017).
- ^{vi} Yoshinaga, N., Nagai, K. H., Sumino, Y. & Kitahata, H. Drift instability in the motion of a fluid droplet with a chemically reactive surface driven by Marangoni flow. *Phys. Rev. E* **86**, 016108 (2012).

REVIEWERS' COMMENTS:

Reviewer #1 (Remarks to the Author):

With respect to my previous review, I commend that the authors have toned down their statements somewhat and clarified some of their argument. However, some bits are still too speculative for publication, with quite a bit of misapplied terminology. However, I regret I have to pass this paper on, I don't have time for another round of reviews before November.

-l 31. comma missing in references.

-l 94. and throughout the MS: 'amplified chirality' referring to molecular chirality transmitted to a mesoscopic LC director variation and subsequently the droplet dynamics. I don't think this is a useful terminology, since it's not that something is getting 'more chiral' here. What's amplified here is the length scale.

-There's still a period in line 97 that is probably a pitch.

-l 119. Again, a bit imprecise. The flow couples to the chiral variation of the director via the anisotropic viscosity, that rotates the internal flow, and then the droplet itself spins because the entire system of droplet and bulk fluid is torque free.

All in all, I don't think the mechanics argument is complete here. To run in a helix, the torque vector T can't be parallel to the velocity vector V , because in that case the droplet would just run straight while spinning. Hence there has to be an additional instability/broken symmetry pulling the torque vector off axis. I refer the authors to Wittkowski et al. Phys. Rev. E 85, 021406 (2012) and Krüger et al. Phys. Rev. Lett. 117, 048003 (2016) for a related phenomenon and the deterministic model.

-The authors should consider splitting fig. 2. Panel e-f are not referenced until much later in the MS.

line 139: it's 'on', not 'from'

-line 142: I would be a bit more careful to directly compare this to E. coli swimming in circles at interfaces. For E. coli, the interface is actually necessary for circling (second broken symmetry again, cf. Lauga et al., Biophys. J. 90, 400–412). Here it's different since the helical swimming is already present. To me it looks like the droplets run in circles because they've run into the wall, and to escape the T/V vectors have to rotate somehow, either due to shear flow at the no-slip wall or from autochemotactic drift (cf. Krüger et al. or Jin, PNAS 114, 5089).

However, wall interactions are not the subject of this study, so the authors could just comment on the fact that they have taken care to only include bulk reorientation events in their analysis. It would still be nice to see the boundaries in the SI. If this is matplotlib3d I refer the authors to eg. <https://stackoverflow.com/questions/23403293/3d-surface-not-transparent-inspite-of-setting-alpha/23413587> for a possible plotting solution :), or to just paint lines on their grids.

I don't understand how the Tuval paper relates to this, it's a fluid dynamics instability driven by bioswimmer collectives and this is a single swimmer problem.

-l 156: I don't think pseudo-nematic is a good term here, since it's commonly used for correlation phenomena in nematogens near the clearing point (cf. Kedziora et al. PRE 66, 031702). Here, the cholesteric pitch just becomes comparable to the droplet dimensions.

-l 165: Why 'bifurcation' instead of 'threshold'? That suggests a transition to two states, e.g. from no chirality to both left and right handed chirality, and from preexisting literature I would assume that the chirality in the system is uniform and fixed by that of the dopant. On that note, can the authors provide the handedness of their helices for completeness in their evaluation?

-l 228: there is no figure 3e.

-l 228-230: This is really confusing. Propulsion doesn't have a radius (do they refer to 'helix') and you can't construct an angle by combining a length and another angle. The droplet doesn't travel 'frequently' away from the helix axis, since this distance is the fixed helix radius R . I assume that

the authors are trying to argue that the angle between V and T (and probably the magnitude of T) increases with N for a reason yet to be determined, which affects both R during steady propulsion and the reorientation angle when T instantaneously switches to -T, or something in that line. I'd again consult Wittkowski et al. on terminology.

-l 250 ff: I'm not sure how much one can actually infer from the BODIPY fluorescence data on the TTAB concentration at the interface. (Orientation dependent quenching? Cosurfactant demixing?) Also, the speculation is a bit shady. Considering the considerable effects of anisotropic viscosity in the droplet bulk fluid, I'd expect them to be dominant over interfacial effects. The authors should also consider the decay time of the viscous flow inside the droplet, which should be slower than an instant chirality inversion due to a local switch in molecular order.

-l 272: I think one of the other reviewers noted that 'spiral droplet' and 'spiral organisation' is imprecise. I concur.

- Conclusions: The authors should stress more what is actually new in this study, especially with respect to refs 37, 41 and 42.

-ref 25: Purcell, not Purcel.

-ref 57: Copy-paste-error: those authors don't end in a.

Reviewer #2 (Remarks to the Author):

The manuscript has been substantially revised and unclear issues have been addressed. I am therefore in favour of the publication of this paper in Nature Communications.

Very minor comment: what are 'empty micelles'? Isn't it sufficient to say 'micelles'?

Reviewer #3 (Remarks to the Author):

The authors have substantially improved the manuscript. In the revised version, all my remarks were taken into account and most of remarks of other two referees. The manuscript is now more clear and more comprehensible to the broad readership of the Nature Communications. Therefore, I support the publication of the paper in the present form in Nature Communications.

Reviewer #1

With respect to my previous review, I commend that the authors have toned down their statements somewhat and clarified some of their argument.

We thank all three reviewers for their critiques and the useful suggestions and for taking time to assess our manuscript.

However, some bits are still too speculative for publication, with quite a bit of misapplied terminology. However, I regret I have to pass this paper on, I don't have time for another round of reviews before November.

We agree with this reviewer our manuscript has become a better articulated paper and we regret that some of the terminology remains confusing to this reader. Both below and in the text, we have addressed every comment by the reviewer, with a special focus on terminology. We trust that our revisions will have resolved most, if not all of the remaining issues.

-l 31. comma missing in references.

Our response: We apologize for this formatting mistake.

-l 94. and throughout the MS: 'amplified chirality' referring to molecular chirality transmitted to a mesoscopic LC director variation and subsequently the droplet dynamics. I don't think this is a useful terminology, since it's not that something is getting 'more chiral' here. What's amplified here is the length scale.

Our response: Reviewer #1 makes a good point and we have thus rephrased throughout the manuscript into “*molecular chirality is transmitted across length scales and eventually into droplet dynamics*”, or along these lines.

-There's still a period in line 97 that is probably a pitch.

Our response: This mistake in terminology has been corrected.

-l 119. Again, a bit imprecise. The flow couples to the chiral variation of the director via the anisotropic viscosity, that rotates the internal flow, and then the droplet itself spins because the entire system of droplet and bulk fluid is torque free. All in all, I don't think the mechanics argument is complete here. To run in a helix, the torque vector T can't be parallel to the velocity vector V , because in that case the droplet would just run straight while spinning. Hence there has to be an additional instability/broken symmetry pulling the torque vector off axis. I refer the authors to Wittkowski et al. Phys. Rev. E 85, 021406 (2012) and Krüger et al. Phys. Rev. Lett. 117, 048003 (2016) for a related phenomenon and the deterministic model.

Our response: These works are indeed relevant and of interest to our research and we thank Reviewer #1 for bringing them to our attention. We have re-phrased the manuscript by using the terminology used in these physics journals and we are confident that the explanation of the mechanism is now clear and understandable for the broad readership of Nature Communications.

The text reads as follows: “*Here, the droplet is chiral, and the spiral organization of the molecules in the droplet is specifically associated with the formation of spiral disclination lines (Figure 1). The molecules are disorganized in these defect lines, and the viscosity is thus lower compared to the interfacial areas where the molecules are oriented homeotropically - in other words the Marangoni flow is faster along the spiral disclination lines. As the entire system of bulk fluid and droplet is torque free, the internal flow consequently rotates, and then the droplet itself spins along a helical trajectory that is characterized by a pitch P and radius R (Figure 2a,b and Supplementary Movie 1). Overall, the Marangoni flow thus couples to the chiral variation of the director via the viscosity. Symmetry*

arguments are of general relevance for driving helical motion and also used in theoretical models for the helical propulsion of orthotropic particles.”

-The authors should consider splitting fig. 2. Panel e-f are not referenced until much later in the MS.

Our response: We are open to the possibility of splitting Figure 2 upon Editorial request, however we think that such a split would not necessarily benefit to the understanding of the work. Please also note that Panels e-f are referenced later in the manuscript, but they are also referenced right after Panels a-d.

- Line 139: it's 'on', not 'from'

Our response: This part of the text has been re-written

-line 142: I would be a bit more careful to directly compare this to E. coli swimming in circles at interfaces. For E. coli, the interface is actually necessary for circling (second broken symmetry again, cf. Lauga et al., Biophys. J. 90, 400–412). Here it's different since the helical swimming is already present. To me it looks like the droplets run in circles because they've run into the wall, and to escape the T/V vectors have to rotate somehow, either due to shear flow at the no-slip wall or from autochemotactic drift (cf. Krüger et al. or Jin, PNAS 114, 5089). However, wall interactions are not the subject of this study, so the authors could just comment on the fact that they have taken care to only include bulk reorientation events in their analysis. It would still be nice to see the boundaries in the SI. If this is matplotlib3d I refer the authors to eg. <https://stackoverflow.com/questions/23403293/3d-surface-not-transparent-inspite-of-setting-alpha/23413587> for a possible plotting solution :), or to just paint lines on their grids.

Our response: We thank the reviewer for this insight, as we are indeed not primarily familiar with the motile behavior of living organisms. We have rephrased what could be misleading, indicating that the mechanism of trapping at the interface is probably different to the mechanism of trapping in the case of swimming microorganisms. The redraft reads: “*Although the trapping mechanism is likely different, microorganisms swimming close to boundaries also swim in circles at interfaces*”.

Further we found the suggested papers relevant and we have added Lauga et al., Biophys. J. 90, 400–412 to the list of references (ref #51 in the second redraft). Kruger et al. were already cited in the manuscript (Ref #45 in this version).

We have indicated in the manuscript that the interaction with the interface has not been taken in consideration and that we have considered only events in the bulk. We have added: “*Hereafter, trapped trajectories and other sections of 2D trajectories, where droplets interact with the boundaries of the chamber, are excluded from the discussion and from any statistical analysis and the data we discuss concerns exclusively free motion and reorientation.*”.

Our response: We agree with Reviewer #1 that although boundary effects are not the subject of this study, it is worthwhile knowing where the boundaries are, and in our experimental set-up the top and bottom of the chamber are always positioned at the same ordinates, which makes it easy to provide this information. We write: “*The top and bottom of the chambers are always at z values 8100 and 7000 μm , respectively*”.

I don't understand how the Tuval paper relates to this, it's a fluid dynamics instability driven by bioswimmer collectives and this is a single swimmer problem.

Our response: This reference was meant to illustrate the effects of interfaces on motile behavior of living organisms, but indeed this specific paper by Tuval describes collective motile behavior and therefore we have removed this reference.

-l 156: I don't think pseudo-nematic is a good term here, since it's commonly used for correlation phenomena in nematogens near the clearing point (cf. Kedziora et al. PRE 66, 031702). Here, the cholesteric pitch just becomes comparable to the droplet dimensions.

Our response: We believe that pseudo-nematic should be an acceptable term here – it usually refers to a state where the chirality of the system is not expressed at the mesoscopic scale and has been used in earlier works on motor-doped liquid crystals. The other reviewers, who are likely familiar with liquid crystal terminology, did not seem to have issues with this term. However, we now use “*compensated nematic*” which is also a good term.

-l 165: Why 'bifurcation' instead of 'threshold'? That suggests a transition to two states, e.g. from no chirality to both left and right handed chirality, and from preexisting literature I would assume that the chirality in the system is uniform and fixed by that of the dopant. On that note, can the authors provide the handedness of their helices for completeness in their evaluation?

Our response: Our experiments indeed confirm that the chirality of motion is uniform and fixed by that of the chiral dopant. We welcome the terminology suggested by the reviewer and use the term “*threshold*” throughout the manuscript.

-l 228: there is no figure 3e.

Our response: We apologize for this referencing mistake and have corrected this into “Figure 4e”.

-l 228-230: This is really confusing. Propulsion doesn't have a radius (do they refer to 'helix') and you can't construct an angle by combining a length and another angle. The droplet doesn't travel 'frequently' away from the helix axis, since this distance is the fixed helix radius R. I assume that the authors are trying to argue that the angle between V and T (and probably the magnitude of T) increases with N for a reason yet to be determined, which affects both R during steady propulsion and the reorientation angle when T instantaneously switches to -T, or something in that line. I'd again consult Wittkowski et al. on terminology.

Our response: We agree that it is the helical trajectory that has a radius and not the propulsion. It is also true that the droplet does not travel frequently away from the helix axis and based on these relevant comments of the reviewer we decided to re-formulate this paragraph by using the terminology proposed by the reviewer.

The text now reads as follows: “The change in the directionality of the droplets was investigated by manual z-tracking (Figure 4b,c). We found that the droplets reorient in a direction characterized by the angle θ , that defines a cone of probability for the reorientation (Figure 4d). Larger values of the number of spiral turns N, measured at the moment of chiral inversion, are associated with larger reorientation angles θ (Figure 4e). The precession angle between the path of the helical trajectory and the droplet axis also increases with N, and this affects both the radius R and pitch P of the helical trajectory during steady motion (as seen from Figure 2f and Supplementary Figure 11). Hence, we argue that the increase of precession angle with N (Figure 2a) is also key to define the reorientation angle, when the angular velocity reverts instantaneously. This understanding is in line with the key role played by precession angles in the swimming behavior of unicellular organisms.”

-l 250 ff: I'm not sure how much one can actually infer from the BODIPY fluorescence data on the TTAB concentration at the interface. (Orientation dependent quenching? Cosurfactant demixing?) Also, the speculation is a bit shady. Considering the considerable effects of anisotropic viscosity in the droplet bulk fluid, I'd expect them to be dominant over interfacial effects. The authors should also consider the decay time of the viscous flow inside the droplet, which should be slower than an instant chirality inversion due to a local switch in molecular order.

Here Reviewer #1 discusses Supplementary Figure 12, which we use in order to illustrate that the spiral disclination lines modify the surfactant distribution at the droplet-water interface. We understand that this image in itself is not conclusive but we would like to point out that this conclusion was reached by other authors before and in particular we refer to the excellent work of Lisa Tran and coworkers [Tran,

L. et al. Shaping nanoparticle fingerprints at the interface of cholesteric droplets. Sci. Adv. 4, eaat8597 (2018).].

On the other hand, while remaining convinced that the reorganization of the surfactants is of importance in the light-induced reorientation, we also realize that the specific role played by surfactant distribution remains speculative at this point of the research. In the context of a communication, the discussion on whether and how the surfactant distribution affects the movement is likely confusing to the reader and therefore we propose to not discuss that point anymore – as written, and with using the formalism proposed by Reviewer #1, we believe the mechanism we propose is both correct and clear.

-l 272: I think one of the other reviewers noted that 'spiral droplet' and 'spiral organisation' is imprecise. I concur.

Our response: Reviewer #2 commented in an earlier round of revisions that the terminology “*spiral shape*” was not appropriate, because the droplet does not really have a spiral shape – it does have a spiral organization, however the surface corrugation is negligible and the shape is overall sphere.

We agree with this comment made in an earlier round of revisions by Reviewer #2 and the text has been amended accordingly. The director field truly follows a spiral organization in these cholesteric droplets, so we maintain that referring to them as “*spiral droplets*” is correct and helpful to the reader. The other reviewers, who are likely familiar with liquid crystal terminology, did not seem to have issues with this expression – as long as it is clear that the droplets are spheres.

- Conclusions: The authors should stress more what is actually new in this study, especially with respect to refs 37, 41 and 42.

Our response: The essential novelty is that the inversion of handedness of the droplets is associated with a deterministic reorientation in their motion. This could not be shown in any of the above-mentioned works because this motile behavior is essentially related to the possibility of a dynamic conversion in the handedness of the spiral droplets.

In addition, and in complement to these afore-mentioned works, we show that the motile behavior of spiral droplets is entirely determined by N = the number of spiral turns in the droplet. In earlier work by some of us [Ref #37], this could not be demonstrated because N was varying during the lifetime of the droplets, under the combined effect of i) concentration of the chiral dopant that was not solubilizing together with the liquid crystal, and ii) shrinkage of the droplet over time.

Ref #21, #41, and #42 describe how the fingerprint texture of a cholesteric liquid crystal rotates in hybrid anchoring conditions. We believe that there is not much in common with the present study.

We have rewritten the abstract and conclusion entirely to better highlight the discovery we made and how this discovery is associated with light-induced inversion of handedness.

-ref 25: Purcell, not Purcel.

-ref 57: Copy-paste-error: those authors don't end in a.

These two mistakes have been corrected.

Reviewer #2

The manuscript has been substantially revised and unclear issues have been addressed. I am therefore in favor of the publication of this paper in Nature Communications.

We thank all three reviewers for their critiques and the useful suggestions and for taking time to assess our manuscript. As a consequence, we believe that this manuscript has become a better articulated paper.

Very minor comment: what are 'empty micelles'? Isn't it sufficient to say 'micelles'?

We like to use the term “*empty micelles*” in contrast to “*filled micelles*”. The empty micelles are those removing some of the liquid crystal and at the origin of the Marangoni flow, and the filled micelles are the result of this process. We refer to Ref # 38, in which the authors also employ the term “*empty micelles*” in this context. However, since we are not discussing the filled micelles here, we agree with Reviewer #2 that it makes sense to simply use “*micelles*”.

Reviewer #3

The authors have substantially improved the manuscript. In the revised version, all my remarks were taken into account and most of remarks of other two referees. The manuscript is now more clear and more comprehensible to the broad readership of the Nature Communications. Therefore, I support the publication of the paper in the present form in Nature Communications.

We thank all three reviewers for their critiques and the useful suggestions and for taking time to assess our manuscript. As a consequence, we believe that this manuscript has become a better articulated paper.